# Standardized Complex Gut Microbiomes Influence Fetal Growth, Food Intake, and Adult Body Weight in Outbred Mice

**DOI:** 10.3390/microorganisms11020484

**Published:** 2023-02-15

**Authors:** Christa N. Cheatham, Kevin L. Gustafson, Zachary L. McAdams, Giedre M. Turner, Rebecca A. Dorfmeyer, Aaron C. Ericsson

**Affiliations:** 1Comparative Medicine Program, Department of Veterinary Pathobiology, University of Missouri (MU), Columbia, MO 65201, USA; 2Molecular Pathogenesis and Therapeutics Program, University of Missouri (MU), Columbia, MO 65201, USA; 3Mutant Mouse Resource and Research Center, University of Missouri (MU), Columbia, MO 65201, USA; 4University of Missouri Metagenomics Center, University of Missouri (MU), Columbia, MO 65201, USA

**Keywords:** microbiome, mouse, birth weight, body weight, intake

## Abstract

Obesity places a tremendous burden on individual health and the healthcare system. The gut microbiome (GM) influences host metabolism and behaviors affecting body weight (BW) such as feeding. The GM of mice varies between suppliers and significantly influences BW. We sought to determine whether GM-associated differences in BW are associated with differences in intake, fecal energy loss, or fetal growth. Pair-housed mice colonized with a low or high microbial richness GM were weighed, and the total and BW-adjusted intake were measured at weaning and adulthood. Pups were weighed at birth to determine the effects of the maternal microbiome on fetal growth. Fecal samples were collected to assess the fecal energy loss and to characterize differences in the microbiome. The results showed that supplier-origin microbiomes were associated with profound differences in fetal growth and excessive BW-adjusted differences in intake during adulthood, with no detected difference in fecal energy loss. Agreement between the features of the maternal microbiome associated with increased birth weight here and in recent human studies supports the value of this model to investigate the mechanisms by which the maternal microbiome regulates offspring growth and food intake.

## 1. Introduction

The prevalence of overweight and obesity are increasing worldwide, facilitated by an increasing availability of processed foods of poor nutritive value and increasingly sedentary lifestyles. Moreover, portion size is poorly controlled for many people, resulting in excessive intake beyond that required to maintain homeostasis for their body mass, metabolic rate, and activity level. The gut microbiota is a major component in the pathophysiology of obesity via the regulation of energy harvest [1] and storage [2], diversification and digestion of dietary macromolecules [3,4], and generation of short-chain fatty acids (SCFAs), molecules that serve as an energy source for colonocytes and regulate local gene expression via histone acetylation and deacetylation [5]. Germ-free mice weigh less than age-matched conventionally housed mice, have reduced visceral adiposity [6,7], and are partially resistant to diet-induced obesity and metabolic dysfunction [8,9].

However, the gut microbiome can also influence host metabolism and body weight through the gut–brain axis and its effects on host behavior. The signaling of microbial metabolites through several G protein-coupled receptors expressed on the gut epithelium induces a release of hormones from enteroendocrine cells and vagal impulses, collectively affecting hunger, satiety, and activity levels [10,11]. Similarly, many metabolites are absorbed into peripheral circulation, allowing for the activation of extraintestinal receptors, including those on hypothalamic cells involved in regulation of behavior [12,13]. These circular and pleiotropic mechanisms connecting the gut microbiome and obesity make research challenging. Complicating matters further, there are transgenerational effects with maternal obesity and the maternal gut microbiome, and both influence offspring growth beginning at conception.

Germ-free and antibiotic-depleted mice have provided valuable insights regarding the effects of the presence or absence of gut bacteria on host metabolism and behavior. That said, germ-free mice are also an inherently artificial system with poorly developed gut epithelial architecture and immunity, making the definitive isolation of causative mechanisms difficult. Moreover, the effects of the microbiome on host outcomes may be the result of interactions (or a balance) between multiple members of the microbiome, individually exerting positive and negative influences. There are several approaches to investigate the effect of different complex microbiomes, or select microbes within a complex community, in mouse models. The use of synthetic communities began over half a century ago with simplified mixtures of autochthonous isolates [14,15], and has since progressed to more complex communities [16] and mixtures of human isolates [17]. While these communities do demonstrate cross-feeding and fulfill many of the metabolic functions of the microbiome, they are still undeniably simple and do not include entire phyla present in many specific pathogen-free (SPF) mice. Alternatively, laboratory mice harboring the microbiome of wild mice better recapitulate the immune ontogeny of humans and provide an ostensibly more translational model system [18,19]. These commercially available communities require specialized housing due to the presence of opportunist microbes excluded from most SPF mouse suppliers and research institutions. Whether a synthetic community or wild mouse microbiome, these represent one group and there are no obvious comparators aside from SPF mice.

Among the SPF mice used in biomedical research, there is a tremendous amount of variability [20], largely attributable to differences in the microbiome of mice provided by domestic (U.S.) suppliers [20,21]. Using embryo transfer of the same germplasm into pseudopregnant surrogate dams from the different suppliers of SPF mice, separate outbred colonies of mice with distinct complex microbiomes were generated. To study the influence of the maternal microbiome on offspring growth and voluntary intake in a controlled biological system, outbred CD-1 mice colonized with two different supplier-origin microbiomes were used [22]. This allows for a comparison of complex, naturally occurring microbiomes without any experimental manipulation, differing consistently in terms of richness and composition, in outbred but largely isogenic hosts, housed under identical conditions. Additionally, previous studies revealed consistent microbiome-associated differences in body weight (BW), anxiety-related behaviors, and locomotor activity between sex- and age-matched mice from these colonies [23].

To determine whether the reproducible GM-associated difference in BW between age- and sex-matched mice is associated with differences in total daily food intake, body weights and daily intake (three consecutive days) were determined for cages of male and female mice colonized with either of the two microbiomes at three, six, and nine weeks of age. To assess the possible contribution of fecal energy loss, fecal samples collected at six weeks of age were subjected to bomb calorimetry. Lastly, to assess inherent differences in growth rate, birth weights were collected from multiple size-matched litters of neonates born to dams colonized by GM1 or GM4. Collectively, our results revealed that the greater BW in age- and sex-matched CD-1 mice colonized with the less rich Jackson-origin GM1 is a result of both greater birth weight and relatively greater intake than age- and sex-matched mice colonized with GM4, even when adjusted for BW. In the context of no detectable difference in fecal energy loss, these findings suggest that the microbiome-mediated effect on BW resulted from both intrinsic effects on growth rate and behavioral components related to the regulation of satiety.

## 2. Materials and Methods

### 2.1. Ethics Statement

This study was conducted in accordance with the recommendations set by the Guide for the Care and Use of Laboratory Animals and was approved by the University of Missouri institutional Animal Care and Use Committee (MU IACUC protocol#36781).

### 2.2. Mice

Six CD1GM1 and eight CD1GM4 (MU Mutant Mouse Resource and Research Center (MMRRC), Columbia, MO, USA) mice were obtained at 9 weeks of age and set up in same GM profile breeding pairs. These mice were produced by existing colonies of mice generated several years ago and maintained as genetically similar outbred colonies of mice, differing in that each colony harbors a distinct supplier-origin microbiome, with GM1 originating from C57BL/6J mice (Jackson Laboratory, Bar Harbor, ME, USA) and GM4 originating from C57BL/6NHsd mice (Envigo, Indianapolis, IN, USA). In summary, these colonies were generated via the embryo transfer of CD-1 germplasm into pseudopregnant C57BL/6J and C57BL/6NHsd surrogate dams. Pups born to those dams acquired their respective supplier-origin microbiomes and served as founders for outbred colonies that have been maintained at the MU MMRRC since their initial description [22,24]. Outbred status was maintained via careful rotational breeding and the annual introduction of new genetic stock from the CD-1 vendor via the embryo transfer of a newly acquired CD-1 germplasm into surrogate dams from the existing colonies. Additionally, the microbiome of these colonies was monitored on a quarterly basis via random sampling of 10 cages of adult breeding trios per colony and 16S rRNA amplicon sequencing of fecal DNA. Three litters per microbiome were culled down to eight mice at birth (with a goal of four male and four female pups) to reduce the possible effects of litter size and differential maternal care on preweaning growth. Due to difficulties in accurately sexing neonatal pups, a total of 20 GM1 offspring (10 cages, 5 male, and 5 female) and 22 GM4 offspring (11 cages, 5 male, and 6 female) were available for weaning into same-sex pairs at three weeks of age. Coprophagy leads to a shared cage-level microbiome and the single housing of mice is nonstandard husbandry that is considered stressful to mice. For these reasons, all adult experimental outcomes were based on the cage (i.e., mouse pair) as the experimental unit in an effort to eliminate cage effects and allow for normal activity and feeding behaviors. All mice were housed under barrier conditions in microisolator cages on ventilated racks with pelleted paper bedding and nestlets as enrichment. All mice had ad libitum access to an irradiated diet (LabDiet 5058 for breeder animals and LabDiet 5053 for feed study animals, LabDiet, St. Louis, MO, USA) and autoclaved tap water under a 12:12 light/dark cycle. The mice were determined to be free from opportunistic bacterial pathogens including *Bordetella bronchiseptica*; cilia-associated respiratory (CAR) bacillus; *Citrobacter rodentium*; *Clostridium piliforme*; *Corynebacterium bovis*; *Corynebacterium kutscheri*; *Helicobacter* spp.; *Mycoplasma* spp.; *Pasteurella pneumotropica*; *Pneumocystis carinii*; *Salmonella* spp.; *Streptobacillus moniliformis*; *Streptococcus pneumoniae*; adventitious viruses including H1, Hantaan, KRV, LCMV, MAD1, MNV, PVM, RCV/SDAV, REO3, RMV, RPV, RTV, and Sendai viruses; intestinal protozoa including *Spironucleus muris*, *Giardia muris*, *Entamoeba muris*, trichomonads, and other large intestinal flagellates and amoebae; intestinal parasites including pinworms and tapeworms; and external parasites including all species of lice and mites via quarterly sentinel testing performed by IDEXX BioAnalytics (Columbia, MO, USA).

### 2.3. Fecal DNA Extraction

At nine weeks of age, two freshly evacuated fecal pellets were collected from each same-sex pair into a 1.5 mL Eppendorf tube. Fecal collections were routinely performed at 6 a.m. DNA was extracted using QIAamp PowerFecal Pro DNA kits (Qiagen, Venlo, The Netherlands) according to the manufacturer’s instructions, with the exception that samples were homogenized in bead tubes using a TissueLyser II (Qiagen, Venlo, The Netherlands) for 10 min at 30/sec rather than the vortex adapter described in the protocol before proceeding according to the protocol, and the DNA was eluted in 100 µL of EB buffer (Qiagen, Venlo, The Netherlands). The DNA yields were quantified via fluorometry (Qubit 2.0, Invitrogen, Carlsbad, CA, USA) using quant-iT BR dsDNA reagent kits (Invitrogen, Waltham, MA, USA).

### 2.4. 16S rRNA Amplicon Library Preparation and Sequencing

The amplicon library preparation and sequencing occurred at the University of Missouri Genomics Technology Core facility. Bacterial 16S rRNA amplicons were constructed via amplification of the V4 region of the 16S rRNA gene with universal primers (U515F/806R) [25,26,27] flanked by Illumina standard adapter sequences. Dual-indexed forward and reverse primers were used in all reactions. A PCR was performed in 50 µL reactions containing 100 ng metagenomic DNA, primers (0.2 µM each), dNTPs (200 µM each), and Phusion high-fidelity DNA polymerase (1U, Thermo Fisher, Waltham, MA, USA). The amplification parameters were 98 °C^(3 min)^ + [98 °C^(15 s)^ + 50 °C^(30 s)^ + 72 °C^(30 s)^] × 25 cycles + 72 °C^(7 min)^. The amplicon pools were combined, mixed, and purified using Axygen Axyprep MagPCR clean-up beads for 15 min at room temperature. The products were washed multiple times with 80% ethanol and the dried pellet was resuspended in 32.5 µL of the EB buffer (Qiagen, Venlo, The Netherlands), incubated for two minutes at room temperature, and then placed on a magnetic stand for five minutes. The amplicon pool was evaluated using an Advanced Analytical Fragment Analyzer automated electrophoresis system, quantified using quant-iT HS dsDNA reagent kits, and diluted according to the Illumina standard protocol for sequencing as 2 × 250 bp paired-end reads on the MiSeq instrument.

### 2.5. Informatics Analysis

The primers were designed to match the 5′ ends of the forward and reverse reads. Cutadapt [28] (version 2.6; https://github.com/marcelm/cutadapt (accessed on 31 October 2022) was used to remove the primer from the 5′ end of the forward read. If found, the reverse complement of the primer to the reverse read was then removed from the forward read as were all bases downstream. Thus, a forward read could be trimmed at both ends if the insert was shorter than the amplicon length. The same approach was used on the reverse read, but with the primers in the opposite roles. Read pairs were rejected if one read or the other did not match a 5′ primer, and an error rate of 0.1 was allowed. Two passes were made over each read to ensure the removal of the second primer. A minimal overlap of three bp with the 3′ end of the primer sequence was required for removal.

The QIIME2 [29] DADA2 [30] plugin (version 1.10.0) was used to denoise, dereplicate, and count ASVs (amplicon sequence variants), incorporating the following parameters: (1) forward and reverse reads were truncated to 150 bases, (2) forward and reverse reads with the number of expected errors higher than 2.0 were discarded, and (3) Chimeras were detected using the “consensus” method and removed. R version 3.5.1 and Biom version 2.1.7 were used in QIIME2. Taxonomies were assigned to final sequences using the Silva.v138 [31,32] database, using the classify-sklearn procedure [33].

### 2.6. Body Weights

At three weeks of age, mice from the same GM background were weaned into same-sex pairs. Their body weights were collected as a pair at roughly 7 a.m. on the day they were weaned. The pair of mice were placed in an empty microisolator cage that was then placed on a tared scale to obtain the combined weight of the pair of mice. This was repeated at exactly six weeks of age as well as at nine weeks of age. To assess the birth weights, breeding pairs and trios were set up and monitored daily for pups. Pups were then sexed and weighed individually within 24 h of birth, and the litter size and sex distribution were recorded. The pup weights were collected using a calibrated NewClassic MF from Mettler Toledo (Columbus, OH, USA). Weaning, six-week, and nine-week weights were obtained using a calibrated Ranger^TM^ 3000 from OHAUS (Parsippany, NJ, USA).

### 2.7. Feed Intake Assessments

At three weeks of age, after the mice were weaned into same-sex pairs, the food compartment of the wire hopper was filled with LabDiet 5053 pellets. The wire hopper with the feed (and no water bottle) was weighed. The wire hopper (with remaining feed) was then weighed daily in this manner for three subsequent days to obtain four consecutive hopper weights and three 24 h differences. The hoppers and feed were weighed at roughly 7 a.m. daily. The three consecutive daily differences for each cage were averaged to obtain the mean daily intake for each cage at three weeks of age. This was repeated again at six weeks of age and at nine weeks of age. The hopper weights were obtained using a Ranger^TM^ 3000 from OHAUS (Parsippany, NJ, USA).

### 2.8. Bomb Calorimetry

At six weeks of age, at least 200 mg of freshly collected feces from both mice in three cages per GM were obtained and placed in an empty Eppendorf tube. The collection occurred at 6 a.m. A total of six samples (three from GM1 and three from GM4) were submitted to the Cornell University Mouse Metabolic Phenotyping Center where the samples were dehydrated for 48 h, and oxygen bomb calorimetry was performed on technical triplicates using a Parr 6765 Combination Calorimeter (Parr Instrument Co., Moline, IL, USA).

### 2.9. Statistical Analysis

Differences in body weight and intake were identified using a two-factor analysis of variance (ANOVA) within a timepoint or a three-factor ANOVA including the timepoint as a variable. Post hoc pairwise comparisons were performed using the Holm Sidak method. To identify substrain-dependent differences in the initial bacterial richness (prior to treatment), non-normal data (as indicated via the Shapiro–Wilk method) were compared via a Mann–Whitney rank sum test. To identify time-dependent differences in richness and alpha diversity, the data were first tested for normality and equal variance via the Shapiro–Wilk and Brown–Forsythe methods, respectively, and then tested via the appropriate parametric or nonparametric two-factor ANOVA. Differences in the relative abundance of features (amplicon sequence variants, ASVs) were detected via parallel analysis using ALDEx2 [34] and ANCOM-BC [35]. This combinatorial approach was selected due to recognized discrepancies among the various differential abundance tools available and a high level of agreement between these two tools and applicability across a wide range of datasets [36]. Differences in beta diversity were detected using a two-way permutational multivariate analysis of variance (PERMANOVA) [37]. Using the vegan package within R v4.2.2 [38], a Bray–Curtis weighted distance matrix of quarter-root transformed features was generated. The two-way PERMANOVA was performed using the *adonis* tool with 9999 permutations. A principal coordinate analysis (PCoA) was performed using the *ape* library [39] with a Calliez correction.

## 3. Results

### 3.1. Supplier-Origin Microbiomes Are Associated with a Difference in Body Weight

As observed in previous cohorts, mice colonized with GM1 weighed more than age- and sex-matched mice colonized with GM4 at weaning (*p* = 0.002, F = 13.6) with no effect of sex (Figure 1A). Post hoc pairwise comparisons indicated significant GM-associated differences in both females (17.42 vs. 15.17 g/mouse; *p* = 0.017, t = 2.6) and males (18.36 vs. 16.11 g/mouse; *p* = 0.02, t = 2.6). The GM-associated difference in body weight (BW) persisted at six weeks of age (31.12 vs. 27.11 g/female mouse, 37.06 vs. 31.36 g/male mouse; *p* = 0.01, F = 8.3) and a sex-associated difference was also detected (*p* = 0.02, F = 6.8), most prominently within the GM1-colonized mice (Figure 1B). Pairwise comparisons detected a significant sex effect within GM1 (*p* = 0.022, t = 2.5) and a significant GM effect within males (*p* = 0.015, t = 2.7). By nine weeks of age, the GM-associated difference in BW (36.10 vs. 29.15 g/female mouse, 42.38 vs. 36.78 g/male mouse; *p* < 0.001, F = 16.3) was comparable to the strong sex bias (*p* < 0.001, F = 25.4; Figure 1C). Accordingly, pairwise comparisons revealed significant effects of GM within females (*p* = 0.003, t = 3.4) and males (*p* = 0.035, t = 2.3) and of sex within GM1 (*p* = 0.008, 3.0) and GM4 (*p* < 0.001, t = 4.2).

### 3.2. Supplier-Origin Microbiomes Are Associated with a Difference in BW-Adjusted Food Intake

Daily food intake was assessed in each cage at three, six, and nine weeks of age using the average difference in hopper weight over four consecutive days at each age. Notably, as early as weaning at three weeks of age, mice colonized with GM1 consumed a greater amount of food than age- and sex-matched mice colonized with GM4 on a per-cage basis (*p* < 0.001, F = 21.2; Appendix A). This difference was significant in both females (*p* < 0.001, t = 4.2) and males (*p* = 0.034, t = 2.3). Similarly, strong GM-associated differences in total intake were observed at six weeks (*p* < 0.001, F = 43.4; Appendix A) and nine weeks (*p* < 0.001, F = 15.6; Appendix A) of age. Interestingly, no differences in total intake were detected between the male and female mice at any age, and no significant GM × sex interactions were detected. When normalized to BW however, the difference in the total intake detected at weaning was abrogated, and no GM- or sex-associated differences in BW-adjusted intake were detected (Figure 2A). At six weeks of age, the BW-adjusted intake was greater in GM-colonized female (*p* = 0.002, t = 3.7) and male (*p* = 0.033, t = 2.3) mice (*p* < 0.001, F = 18.2; Figure 2B). The BW-adjusted intake was also greater in the GM1-colonized mice at nine weeks of age (*p* = 0.011, F = 8.0; Figure 2C), and pairwise comparisons indicated that this GM-associated difference was significant in females (*p* = 0.015, t = 2.7) but not males (*p* = 0.199, t = 1.3) at this timepoint.

### 3.3. Mice Colonized with GM4 Reduce BW-Adjusted Intake over Time

A comparison of all three datasets over time revealed several interesting patterns. The body weights suggested comparable growth curves in all groups and greater growth from three to six weeks of age than from six to nine weeks (Figure 3A). While the three-way ANOVA confirmed age as the dominant factor in BW, significant (and comparable) GM- and sex-dependent effects were also observed, with only modest interactions between any of the variables. The total intake and BW-adjusted intake showed complementary trends over time. Specifically, mice colonized with GM1 consumed a greater total amount of food at each successive timepoint (3 w vs. 6 w, *p* < 0.001, t = 4.2; 6 w vs. 9 w, *p* = 0.069, t = 1.9) while no differences between timepoints were detected in total intake by mice colonized with GM4 (Figure 3B). In contrast, the BW-adjusted intake was consistent across time in the GM1-colonized mice but was significantly reduced at six weeks (*p* < 0.001, t = 4.8) and nine weeks (*p* < 0.001, t = 5.0) of age, compared to at three weeks of age. No difference in the BW-adjusted intake was detected in the GM4-colonized mice between six and nine weeks of age.

### 3.4. Supplier-Origin Microbiomes Are Associated with a Difference in Fetal Growth

Due to the differences in BW and total daily intake (but not BW-adjusted daily intake) present at weaning, we reasoned that mice colonized with GM1 may be programmed for a faster rate of growth at birth. To assess this, birth weights (at less than 24 h of age) were assessed in 119 pups from five and six litters born to dams colonized with GM1 or GM4, respectively. While there was no apparent relationship between litter size and birth weight (Appendix A), analyses were performed in parallel using four litters per microbiome of a roughly equivalent size distribution and all 11 litters. A comparison of the four litters selected from each microbiome revealed no difference in litter size (*p* = 0.82, *t*-test, Figure 4A) or sex distribution (*p* = 0.25, χ^2^-test, Figure 4B) but a profound GM-dependent difference in birth weight (*p* < 0.001, F = 139.9, Figure 4C). There was a trend (*p* = 0.06, F = 3.8) toward a sex-associated difference and no interaction between GM and sex. The analysis using all 11 litters yielded identical results (Appendix A), showing quite convincingly that pups born to dams colonized with GM1 are larger than pups born to dams colonized with GM4.

### 3.5. Supplier-Origin Microbiomes Differ in Composition but Not in Fecal Energy Loss

Lastly, to assess the fecal energy loss and confirm the previously observed compositional differences between GM1 and GM4, fecal samples were collected at six weeks of age for bomb calorimetry and at the endpoint (nine weeks) for 16S rRNA amplicon sequencing. As previously reported, GM-associated differences in richness (*p* = 3.3 × 10^−7^, F = 61.4; Figure 5A) and Shannon alpha diversity (*p* = 1.4 × 10^−4^, F = 23.2; Figure 5B) were detected with no significant effect of sex or GM × sex interactions in either metric. Similarly, differences in beta diversity were detected via two-way PERMANOVA (*p* = 1.0 × 10^−4^, F = 12.1; Figure 5C). Testing for the differential abundance (DA) of genera, performed via parallel analysis with ALDEx2 and ANCOM-BC, resulted in the identification of 32 differentially abundant genera with a Benjamini–Hochberg-corrected [40] *p* value < 0.05 detected with both tools (Appendix A). The genera found to be enriched in GM1 included ten from the phylum *Bacillota*, including *Ruminococcus*, *Roseburia*, and *Butyricicoccus*; two from the *Mycoplasmatota* (RF39 and *Anaeroplasma*); and one each from the *Bacteroidota* (unresolved *Tannerellaceae*), *Saccharibacteria* (*Candidatus Saccharimonas*), *Actinomycetota* (*Gordonibacter*), and *Verrucomicrobiota* (*Akkermansia*). The genera enriched in GM4 included eight *Bacillota*, four *Bacteroidota* including two *Rikenellaceae*, two sulfate-reducing bacteria (*Desulfovibrio* and *Bilophila*), and members of two other bacterial phyla. A comprehensive list of taxa identified as DA by ANCOM-BC based on structural zeros in either group is provided in Appendix A. The same DA analyses performed to identify sex-dependent differences in the abundance of taxa identified no differences in the relative abundance of any taxa between females and males. No difference was detected in the fecal energy content in the samples collected from six-week-old females and assayed in duplicate (Appendix A).

## 4. Discussion

GM1 and GM4 represent two distinct, complex, naturally occurring specific pathogen-free (SPF) mouse gut microbiomes. They were used here in two large colonies of outbred mice as a means of modeling microbiome-dependent host phenotypes occurring within humans at the population level [22]. There are four primary producers of SPF mice in the U.S.: the Jackson Laboratory, Taconic, Charles River Labs, and Envigo. As the names GM1 and GM4 imply, our lab previously maintained four separate colonies, each harboring a microbiome originating from those four producers. The microbiomes were numbered GM1 through GM4 based on the order of increasing richness. In addition to differences in richness, GM1 and GM4 also differed most in terms of beta diversity. The mice with GM1 (or GM2) were the heaviest and showed the greatest anxiety-related behavior, while the mice with GM4 were consistently the leanest, most active, and showed the least anxiety-related behavior, leading to focused investigations of GM1 and GM4 and the exclusion of the two microbiomes that were intermediate in terms of richness, beta diversity, and phenotypic outcomes.

Previous and ongoing work has revealed numerous GM-associated differences in these colonies in terms of anxiety-related behavior, voluntary locomotor and exploratory activity, and growth [23]. Specifically, mice colonized with GM1 are typically heavier, more anxious, and less active than age- and sex-matched mice colonized with GM4. While the greater activity may explain, at least partially, the difference in BW, we hypothesized that differences in other behaviors, such as feeding behaviors, might also contribute to the difference in BW. The current data suggest that while the mice colonized with GM1 did indeed consume more food than age- and sex-matched mice colonized with GM4, these mice were born heavier and there was no difference in BW-adjusted intake at early time-points. By adulthood however, the mice colonized with GM1 tended to increase their total daily intake beyond that ostensibly necessary to account for the difference in physiological demand. Several microbially derived metabolites, including unconjugated bile acids (BAs) and short chain fatty acids (SCFAs), are capable of binding receptors on enteroendocrine cells (EECs) and inducing the activation of vagal afferents [41] and the expression of neuroactive hormones associated with satiety such as glucagon-like peptide 1 (*Glp1*) and peptide YY (*Pyy*) [42,43,44]. As gut bacteria differ widely in their ability to deconjugate specific BAs, synthesize specific SCFAs, and produce a myriad other biologically active molecules, it is reasonable to speculate that this variegated functional capacity of the gut microbiome results in some cognate mixture of metabolites and host responses in EECs and the CNS, ultimately affecting hunger, satiety, and feeding behavior.

Notably, we believe that the GM-associated differences in BW reflect a difference in growth rather than adiposity. While we have previously reported consistent differences in BW between age- and sex-matched mice colonized with GM1 or GM4, mice also differ in body length and cardiac weight at 16 weeks of age, neither of which remain different when normalized to overall body weight [23]. Moreover, DEXA scanning of 14-week-old mice showed no difference in body composition, collectively suggesting that mice colonized with GM1 are simply growing faster than sex-matched mice with GM4 [23]. The observed difference in birth weight in the current study indicates a faster rate of fetal growth in the GM1 dams and supports this model. The bidirectional link between the gut microbiome and endocrine factors controlling growth is an active area of investigation relevant to both pediatric medicine and livestock production [45,46,47].

Speaking to the potential translatability of these findings, two recent studies in humans have identified significant associations between features of the maternal gut microbiome prior to or during pregnancy and infant birth weights. A recently published study of women in their first trimester of pregnancy identified a negative correlation between alpha diversity indices and infant birth weights [48], as was observed in the present study. Similarly, a study in a much larger cohort of pregnant women from Zimbabwe found that specific taxonomic or metagenomic features of the pregnant microbiome were better predictors of infant birth weight than gestational age [49]. In particular, the taxa and KEGG Ortholog (KO) gene categories associated with SCFA production were positively correlated with birth weight [49]. We note that taxa enriched in GM1 included several butyrogenic families such as *Lachnospiraceae*, *Ruminococcaceae*, and *Oscillospiraceae* and the acetogenic RF39 group [50]. The similarity between these studies and the features found in GM1 suggest the possibility of a common mechanism and support the utility of the mouse model used in these studies. While a lower microbial richness was associated with greater BW in the current study, the germ-free (GF) mice were typically leaner than the SPF mice and were relatively resistant to diet-induced obesity [8]. This suggests that the effects of the microbiome on BW are not necessarily related to richness per se but rather characteristic features within the microbiome. Additionally, while the ceca of the GF mice were dramatically enlarged compared to that of the SPF mice, there were no grossly visible differences in cecal size or appearance between the mice colonized with GM1 or GM4.

The lack of a detected difference in fecal energy loss was not surprising given the efficiency of the microbial energy harvest. This was a small sample size, and we did not measure the 24-h fecal output, limiting our interpretation of the data. However, the high intra- and intersample agreement despite the observed differences in weight and intake at the same time point suggest that differences in fecal energy loss contribute little, if any, to the consistent GM-associated differences in BW and growth. Additionally, we have not assessed the basal metabolic rate (BMR) in mice colonized with GM1 or GM4, or the influence of the maternal gut microbiome on offspring BMR.

Another limitation of the current work is the method used to measure intake. While an accepted method [51], differences in food hopper weight may falsely inflate estimations of intake due to food crumbs that are dropped into the bedding and not consumed. That being said, no differences in food crumbs were observed beneath the wire hoppers when we obtained the hopper weights.

Collectively, the current data suggest that a greater growth rate is favored by the low richness GM1, and that this effect begins during fetal development in response to the maternal microbiome. Moreover, our data suggest that while the larger mice colonized with GM1 initially consumed a comparable BW-adjusted amount of food compared to mice colonized with GM4, this difference eventually transitioned to excessive intake. Considering the lack of difference in fecal energy loss, we speculate that the GM-associated differences in adult BW were due in part to behavioral differences including intake and voluntary activity [23], but also due to inherent differences in growth beginning during fetal development.

## Figures and Tables

**Figure 1 microorganisms-11-00484-f001:**
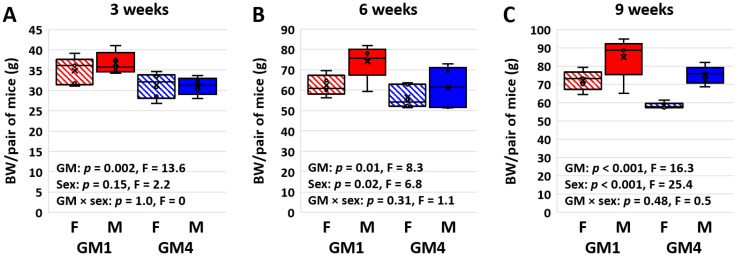
Box plots showing the collective body weight (BW) of pairs of female (F) or male (M) mice colonized with GM1 or GM4 at three (**A**), six (**B**), and nine (**C**) weeks of age (*n* = 5–6 cages/sex/GM). Statistical analysis performed using two-way ANOVA within each time point.

**Figure 2 microorganisms-11-00484-f002:**
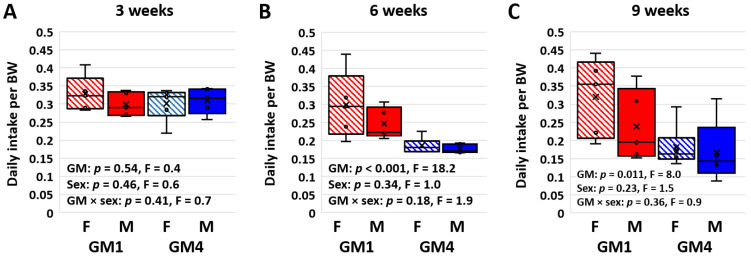
Box plots showing the body weight (BW)-adjusted daily intake in cages housing pairs of female (F) or male (M) mice colonized with GM1 or GM4 at three (**A**), six (**B**), and nine (**C**) weeks of age (*n* = 5–6 cages/sex/GM). Statistical analysis was performed using two-way ANOVA within each time point.

**Figure 3 microorganisms-11-00484-f003:**
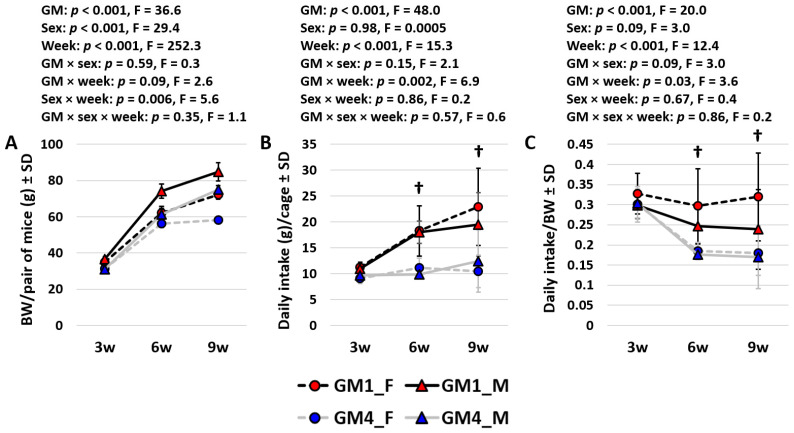
Line plots showing the mean (±SD) body weight (BW) (**A**), total daily intake (**B**), and BW-adjusted daily intake (**C**) in cages housing pairs of female (F) or male (M) mice colonized with GM1 or GM4 at three (3 w), six (6 w), and nine weeks (9 w) of age (*n* = 5–6 cages/sex/GM). Statistical analysis was performed using three-way ANOVA. Daggers indicate significant GM-dependent differences in post hoc comparisons.

**Figure 4 microorganisms-11-00484-f004:**
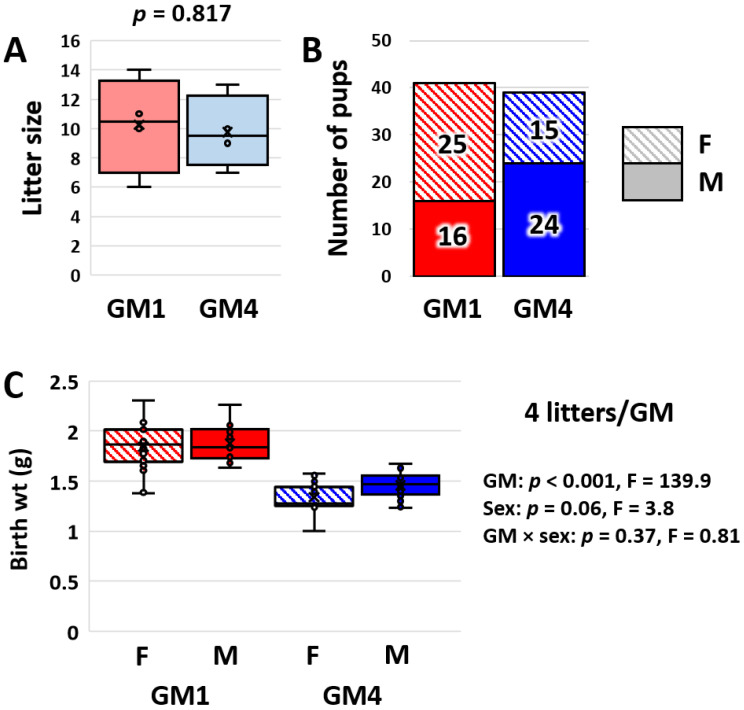
Box and bar plots showing the distribution of litter sizes (**A**) and sexes (**B**) in four litters of roughly equivalent sizes born to dams colonized by GM1 or GM4, and box plot showing birth weights of female (F) and male (M) mouse pups born to GM1- or GM4-colonized dams (**C**). Results of two-way ANOVA on right.

**Figure 5 microorganisms-11-00484-f005:**
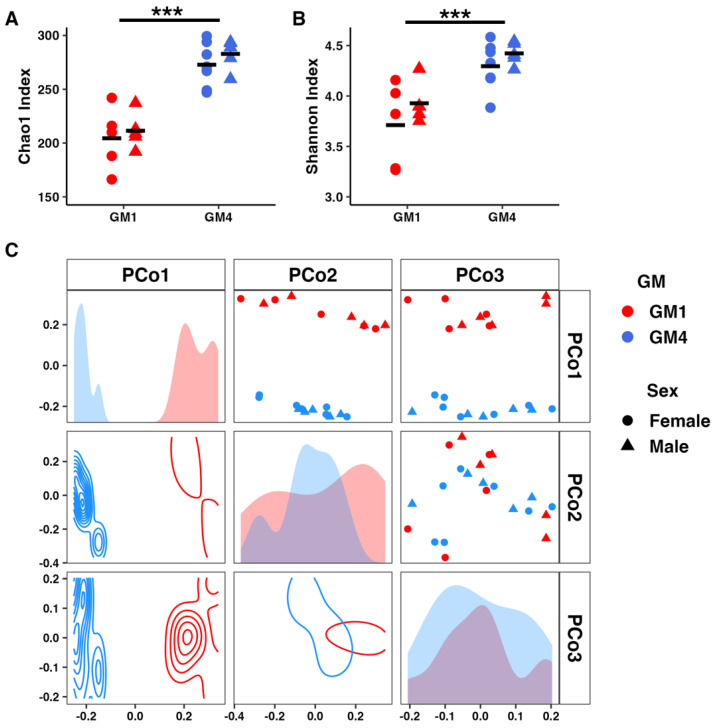
Dot plots showing detected richness (**A**) and Shannon diversity (**B**) across the cages used in the intake study, and principal coordinate analysis (PCoA) matrix depicting microbial community structure along the first three principal coordinates using Bray–Curtis distances (**C**). Dot plots depict individual samples along two axes. Density plots depict sample distribution along two axes. Histograms depict sample distribution along a single axis. Triple asterisk indicates *p* < 0.001.

## Data Availability

All 16S rRNA amplicon sequencing data supporting the current manuscript have been deposited in the National Center for Biotechnology Information (NCBI) Sequence Read Archive (SRA) under the same title as the current project, and BioProject ID PRJNA918350.

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
