# Peer review of "Standardized Complex Gut Microbiomes Influence Fetal Growth, Food Intake, and Adult Body Weight in Outbred Mice"

_microorganisms, 2023, doi:10.3390/microorganisms11020484_

Round 1

Reviewer 1 Report

1. From the "Materials and Methods", I can not know how authors design the experiment to explain "Standardized complex gut microbiomes".

2. from "2.2. Mice", I can not  know how the group was divided.

3.Why was done "At 9 weeks of age, two freshly evacuated fecal pellets were collected between 6 and 101 8 a.m", and not at same time?

4. I don't know what is “2.416. S rRNA”.

5. The samples were collected at different times, so the results were not believed.

6. It's not scientific for "three from GM1 169 and three from GM4 ".

7. How to explain the relation of "Standardized complex gut microbiomes influence fetal growth, 2 food intake, and adult body weight in outbred mice“?

8. Many spelling errors.

Author Response

We sincerely appreciate your time and effort to review our manuscript.  Based on the comments of all reviewers, the revised manuscript has been greatly improved and we are grateful for your contribution. We list your numbered comments here, followed by our responses to each in bold text.  Attached is the revised manuscript, and I'm working to submit two new supplemental tables generated in response to reviewer suggestions.

  1. From the "Materials and Methods", I can not know how authors design the experiment to explain "Standardized complex gut microbiomes".

We appreciate the reviewer’s comment and apologize for the lack of clarity regarding the mice and microbiomes used in this study.  The two microbiomes tested in the current study represent standardized, complex bacterial communities.  The NIH U42-funded Mutant Mouse Resource and Research Center (MMRRC) generated these colonies of mice via embryo transfer in 2016, to study the influence of different specific pathogen-free (SPF) microbiomes occurring in mice supplied by different domestic (U.S.) vendors.  The two microbiomes in question (GM1 and GM4) are both complex, naturally occurring SPF microbiomes that have now been maintained in two genetically identical colonies of mice, with careful breeding and introduction of new stock to maintain outbred status in each colony, and quarterly monitoring of fecal DNA via 16S rRNA amplicon sequencing. We previously demonstrated reproducible effects of these two microbiomes on anxiety-related behavior, locomotor activity, and weight gain in these mouse colonies (reference 23 in initial submission). The purpose of the current studies was to assess microbiome-associated differences in weight-adjusted food intake, fecal energy loss, and fetal growth in age-, sex-, and genotype-matched mice. We have expanded section 2.2. Mice substantially, to include all of these relevant details regarding creation and maintenance of these colonies.

  1. from "2.2. Mice", I can not  know how the group was divided.

Similarly, we apologize for the lack of clarity regarding how mice were generated for the intake studies. As above, section 2.2. Mice has been expanded to make these details clear.

3.Why was done "At 9 weeks of age, two freshly evacuated fecal pellets were collected between 6 and 101 8 a.m", and not at same time?

We routinely began collecting hopper and body weights at the same time each day, and listed a time range simply because we began at the same time each day, but it often took between 30 and 60 minutes to finish collecting all data.  For the sake of clarity, we have amended the text to simply reflect the time at which sample collection began each day.

  1. I don't know what is “2.416. S rRNA”.

This appears to be the result of a type-setting error, causing ‘16S rRNA’ to be altered.  We appreciate the reviewer’s careful reading and have corrected this in the revised manuscript (line 152).

  1. The samples were collected at different times, so the results were not believed.

As clarified above, samples were collected at the same time for both groups, and at the same time each day.  This has been clarified in the manuscript.

  1. It's not scientific for "three from GM1 169 and three from GM4 ".

We agree that this was a low-powered experiment, and this is a limitation of the bomb calorimetry data.  These assays were very cost-prohibitive, leading us to submit three pooled samples per group to determine whether testing of a larger sample size was warranted.  Based on the almost identical mean values obtained with n = 3/group, it was difficult to justify the additional investment of limited funds.  We fully acknowledge these limitations in the Discussion, although I am highly doubtful that analysis of more samples would reveal a difference in fecal energy loss.

  1. How to explain the relation of "Standardized complex gut microbiomes influence fetal growth, 2 food intake, and adult body weight in outbred mice“?

Mechanistically, our data provide strong evidence that differences in adult body weight (BW) are, at least partially, explained by differences in fetal growth (BW at birth) and food intake (across time, adjusted for BW).  In other words, these findings support both inherent growth-associated and behavioral components of the highly reproducible microbiome-dependent difference in adult BW. We apologize if we have misinterpreted the reviewer’s comment.

  1. Many spelling errors.

We have reviewed the manuscript very carefully, and fixed all identified typographical errors.  We note that there are several words identified by SpellCheck as misspellings, that are actually trade names (e.g., PowerFecal, TissueLyser).  Aside from those, we have reviewed the manuscript carefully and identified no other spelling errors.

Reviewer 2 Report

The aim of this study is to determine whether GM-associated differences in BW are associated with differences in intake fecal energy loss, or fetal growth. The authors found that supplier-origin microbiomes are associated with profound differences in fetal growth, and excessive BW-adjusted differences in intake during adulthood, with no detected difference in fecal energy loss. It is an interesting research. However, it may be beneficial for the authors to revise the manuscript to improve clarity and readability, taking into consideration the following suggestions.

1. In the final paragraph of the introduction, the authors are suggested to briefly introduce the research content and significance of this study.

2. In section 2.2. Mice, the authors mention that "Their offspring from these pairs were culled down to 8 mice." It would be helpful for the reader to understand the authors' experimental design if the authors could explain the standards and reasons for the culling.

3. It is suggested that the authors include content on "statistics" in the "2. Materials and Methods" section to ensure the completeness and scientificity of the manuscript.

4. It is suggested that the authors thoroughly discuss the limitations of this study in the "4. Discussion" section and provide a perspective on future research directions.

5. Although I do not feel qualified to judge the English language and style, the manuscript does not read smoothly and there are some minor typographical errors. It is suggested that the authors review and proofread the manuscript to ensure that the writing is coherent and the formatting is correct.

Author Response

We sincerely appreciate your time and effort to review our manuscript.  Based on the comments of all reviewers, the revised manuscript has been greatly improved and we are grateful for your contribution. We list your numbered comments here, followed by our responses to each in bold text. Attached is the revised manuscript, and I am working to upload two new supplemental tables generated based on reviewer suggestions.

The aim of this study is to determine whether GM-associated differences in BW are associated with differences in intake fecal energy loss, or fetal growth. The authors found that supplier-origin microbiomes are associated with profound differences in fetal growth, and excessive BW-adjusted differences in intake during adulthood, with no detected difference in fecal energy loss. It is an interesting research. However, it may be beneficial for the authors to revise the manuscript to improve clarity and readability, taking into consideration the following suggestions.

  1. In the final paragraph of the introduction, the authors are suggested to briefly introduce the research content and significance of this study.

We appreciate the reviewer’s suggestion and have added an additional (final) paragraph to the Introduction, as recommended.

  1. In section 2.2. Mice, the authors mention that "Their offspring from these pairs were culled down to 8 mice." It would be helpful for the reader to understand the authors' experimental design if the authors could explain the standards and reasons for the culling.

Again, this is an excellent suggestion. As CD-1 mice can have litters with up to 20 pups, this was done to minimize differences between litters in competition and access to maternal nutrition and care. Details related to the rationale and relevant experimental methods have been add to section 2.2. Mice (lines 118-121 in revised manuscript)

  1. It is suggested that the authors include content on "statistics" in the "2. Materials and Methods" section to ensure the completeness and scientificity of the manuscript.

We apologies for this oversight.  A detailed description of all statistical methods has been added to the revised manuscript (lines 215 to 234 in revised manuscript)

  1. It is suggested that the authors thoroughly discuss the limitations of this study in the "4. Discussion" section and provide a perspective on future research directions.

Previous limitations included in the original Discussion have been expanded to include limitations related to methods of measuring food intake.

  1. Although I do not feel qualified to judge the English language and style, the manuscript does not read smoothly and there are some minor typographical errors. It is suggested that the authors review and proofread the manuscript to ensure that the writing is coherent and the formatting is correct.

We apologize for any lack of eloquence.  We have reviewed the manuscript closely and corrected grammatical and typographical errors to the best of our ability.

Reviewer 3 Report

Article devote to the effect of the intestinal microbiome of mothers on the weight of offspring and food intake. The work continuant previously published data of authors on mice with the same metabolome where they studied behavior.

The study is interesting but some data in the article is repeated. Metabolomic analysis underrepresented, except for alpha and beta diversity and principal analysis. There are no represented comparison main genus bacteria between two colonies of mice. A previously published article has similar comparisons, but it presents 5 groups. This is important, as there are bacteria that contribute to weight loss and increased metabolism.

It cannot be argued that this is not related to obesity, since there is no direct evidence in this work, as well as in previously published ones. The authors should pay attention to the fact that a decrease in bacterial diversity and a decrease in the number of those bacteria that provide enzymatic activity. A similar effect is observed in germ-free mice, in which the volume and weight of the rectum increases in the absence of bacteria. This should be discussed in this article.

For a better understanding of the data obtained, the authors should change the presentation of the results:

Figure 1 and Figure 2 repeat the graphs shown in Figure 3. In this regard, Figures 1 and 2 should be removed. Rewrite the text also according to these changes.

In the article, the calculation of the animal's weight and feed intake is carried out per cage. it is very difficult to understand. Needs to be counted per mouse. Then the data obtained in the article can be easily compared with similar ones obtained by other researchers.

The method of assay kcal in feces is not described. It is necessary to briefly describe the method, even if it was done by a commercial organization.

Need to indicate from with pathogen and virus mice were free. Line 98-99 “from quarterly testing of sentinel mice done by IDEXX BioAnalytics” If it is FELASA quarterly that need to add year and citation. If it is your profile need to write which bacteria and virus detected

Need to add statistic analysis into Materials and methods

A link to table s1 was provided, but it is not in the supplementary

The supplementary contains figs that are not referenced in the text. What are its needed for?

Insert an fig of the main taxonomic groups of bacteria by which the groups differ from each other.

Line 264 states that "mice colonized with GM1 are typically heavy, more anxious, and less active"

But this was not obtained in this study, and there is no reference to where this data was obtained from?

Include in the discussion about the increase in body weight of the colon and caecum. Or show evidence that this is not due to an increase in the size of the intestine.

Line 94 Change “Chow” to “Diet”

Line 96 Miss dot

Line 109 typo

Line 158 wire hopper Does it bin from wire? Please change to be clear

Author Response

We sincerely appreciate your time and effort to review our manuscript.  Based on the comments of all reviewers, the revised manuscript has been greatly improved and we are grateful for your contribution. We list your numbered comments here, followed by our responses to each in bold text. Attached is the revised manuscript, and I am working to upload two new supplemental tables generated based on reviewer suggestions.

Article devote to the effect of the intestinal microbiome of mothers on the weight of offspring and food intake. The work continuant previously published data of authors on mice with the same metabolome where they studied behavior.

The study is interesting but some data in the article is repeated. Metabolomic analysis underrepresented, except for alpha and beta diversity and principal analysis. There are no represented comparison main genus bacteria between two colonies of mice. A previously published article has similar comparisons, but it presents 5 groups. This is important, as there are bacteria that contribute to weight loss and increased metabolism.

We appreciate the reviewer’s suggestion and have revised Figure 5 to include a PCoA matrix containing a third principal coordinate (visualized three different ways), and added Tables S1 and S2 containing the results of parallel differential abundance testing performed with ALDEx2 and ANCOM-BC. This table provides the information requested by the reviewer in a tabular format to best reflect the degree and directionality of detected differences.  These two particular statistical approaches were selected for their consistent and generalizable results across multiple datasets (as shown by Nearing et al, 2022, Nature Communications, 13:342, cited in the revised manuscript), and used in combination to enhance rigor and reproducibility. Table S1 lists those taxa identified as significantly different by both tools, while Table S2 provides a longer list of ASVs identified as significantly abundant by ANCOM-BC via structural zeros.

It cannot be argued that this is not related to obesity, since there is no direct evidence in this work, as well as in previously published ones. The authors should pay attention to the fact that a decrease in bacterial diversity and a decrease in the number of those bacteria that provide enzymatic activity. A similar effect is observed in germ-free mice, in which the volume and weight of the rectum increases in the absence of bacteria. This should be discussed in this article.

While the reviewer’s comments are noted, we would respectfully submit that our previously published phenotyping of these CD-1 colonies included DEXA scanning of age- and sex-matched mice, which detected no difference in fat mass or lean muscle mass between strains, despite significant differences in BW, body length, and cardiac weight (the latter of which was abrogated when adjusted for BW (cited in the manuscript). Collectively, and in the context of the reported difference in fetal growth, these data suggest that factors, beyond adiposity alone, contribute to the difference in adult BW. Additionally, the greater cecal size in GF mice is due to loss of mucolytic activity, mucin accumulation, and osmotic retention of fluid (Bleich & Hansen, 2012, Comp Immunol Microbiol Inf Dis, 35(2):81).  In fact, GF mice are leaner than feed-matched mice colonized with an SPF microbiome, and several studies have demonstrated resistance of GF mice to diet-induced obesity (Bäckhed et al (2007) PNAS, 104(3): 979).  That being said, it is an interesting consideration and we have added a section to the Discussion describing this thought, and relevant published and anecdotal data.

For a better understanding of the data obtained, the authors should change the presentation of the results:

Figure 1 and Figure 2 repeat the graphs shown in Figure 3. In this regard, Figures 1 and 2 should be removed. Rewrite the text also according to these changes.

While we appreciate the reviewer’s suggestion, we respectfully contend that Figures 1 and 2 provide a better visualization of body weight (BW) and BW-adjusted intake across time and the statistical results of tests within each timepoint, while Figure 3 provides the trends in those values (as well as unadjusted intake) across time.  We believe that each Figure provides important information to readers and would ask that they be retained.

In the article, the calculation of the animal's weight and feed intake is carried out per cage. it is very difficult to understand. Needs to be counted per mouse. Then the data obtained in the article can be easily compared with similar ones obtained by other researchers.

This is not possible as adult weights and daily intake were measured on a cage basis, rather than individually. This was done intentionally due to the inherent coprophagy practiced by mice and subsequent ‘cage effect’ present in fecal microbiome data, as well as the perceived stress of being housed singly. As our goal was to assess normal feeding behavior in unstressed animals, pair housing avoided individual housing. As this necessitated measurements of daily intake on a cage basis, weights were collected similarly, and the cage became the experimental unit for all adult data. The rationale and details of this approach are now described in section 2.2. Mice of the Methods.

The method of assay kcal in feces is not described. It is necessary to briefly describe the method, even if it was done by a commercial organization.

We appreciate the suggestion and have attempted to contact the Center that performed the bomb calorimetry methods (several times), but have been unable to receive a response.  Based on their website, we have added the instrument used to perform the measurements.  Other than that, we have listed all known details, and hope that this is adequate.

Need to indicate from with pathogen and virus mice were free. Line 98-99 “from quarterly testing of sentinel mice done by IDEXX BioAnalytics” If it is FELASA quarterly that need to add year and citation. If it is your profile need to write which bacteria and virus detected

We sincerely appreciate the reviewer’s astute suggestion.  We have added the complete list of tested and excluded pathogens in section 2.2. Mice of the Methods (lines 131 to 141).

Need to add statistic analysis into Materials and methods

We apologies for this oversight.  A detailed description of all statistical methods has been added to the revised manuscript (lines 215 to 234).

A link to table s1 was provided, but it is not in the supplementary

We apologize for this error and have uploaded Table S1 with the revised manuscript.

The supplementary contains figs that are not referenced in the text. What are its needed for?

Upon review, we realized that Figure S1 was not referenced in the text. This has been corrected, and that figure is now cited in Section 3.3.  All other supplementary figures are referenced in the text.

Insert an fig of the main taxonomic groups of bacteria by which the groups differ from each other.

We appreciate the suggestion to provide additional information regarding taxonomic differences between groups.  Rather than an additional figure, we respectfully submit the revised manuscript containing new Tables S1 and S2 listing the full results of differential abundance tests. Recognizing the differences in output and interpretation of commonly used differential abundance tests, we selected two complementary approaches recently were selected for their consistent and generalizable results across multiple datasets (as shown by Nearing et al, 2022, Nature Communications, 13:342, cited in the revised manuscript), and used in combination to enhance rigor and reproducibility. Table S1 lists those taxa identified as significantly different by both tools, while Table S2 provides a longer list of ASVs identified as significantly abundant by ANCOM-BC via structural zeros. We wanted to provide the comprehensive outputs of these tests, resulting in the decision to opt for a tabular, rather than graphic, display of the data.

Line 264 states that "mice colonized with GM1 are typically heavy, more anxious, and less active"

But this was not obtained in this study, and there is no reference to where this data was obtained from?

We apologize for not including the citation.  This sentence was referencing our previous publication.  The reference has now been added.

Include in the discussion about the increase in body weight of the colon and caecum. Or show evidence that this is not due to an increase in the size of the intestine.

We did not collect weight or length measurements of the colon and cecum as we have never observed appreciable differences. Specifically, GM1 and GM4 ceca look similar in size, and studies using B6 substrains colonized with GM1 and GM4 found no difference in colon length of untreated adult mice. As mentioned above, we have added a section in the Discussion related to these questions (lines 388 to 394).

Line 94 Change “Chow” to “Diet”

Line 96 Miss dot

Line 109 typo

We appreciate the reviewer’s careful reading.  Those errors have now been corrected.

Line 158 wire hopper Does it bin from wire? Please change to be clear

Yes, the wire hopper (containing feed) was weighed daily for four consecutive days for each cage, at each timepoint. We have made our best effort to clarify the wording in this section.

Reviewer 4 Report

The manuscript deals with an interesting topic i.e., the role of vendor-specific microbiomes on fetal growth in outbred mice; however, there are multiple shortcomings that make it unsuitable for publication in the current version.

Mice were colonized with the supplier-origin microbiome, but this is only mentioned in the introduction (Line 75-76), not in Material and Methods.

Line 143-144: Authors used SILVA ver. 132 for the taxonomy assignment, but SILVA ver. 138 is available from 2020 and there are huge differences between the two versions. It would be worthwhile to check and update the results with the updated database rather than an outdated and obsolete database.  

Figure 1-2: The authors calculated the p-value for the GM and sex difference, but they did not calculate the p-value for the GM difference in the same sex, and sex difference in the same GM.

Line 175-176: The authors provide results that colonization of GM1 and GM4 generates body weight differences as in the previous study. It has to be discussed precisely in the discussion from the point of view of the microbial content difference of supplier-origin microbiome. To do so, they must measure the microbial composition after colonization and need to link the difference between groups.

Line 189-192: The way of interpreting and describing the results is a bit strange and misleading. The authors can simply describe the results as “When the daily intake was adjusted by weight, there was a large difference between GMs regardless of sex, indicating a strong GM-dependent effect”. However, the meaning is ambiguous due to the use of unnecessary negative sentences like “can no longer be explained…”.

Figure 3: Post-hoc comparison between time points of each group and between GMs within the same time point should be presented for accurate interpretation and understanding.

Line 242-250: The authors did not provide any results regarding bacterial abundance and colonization results. Even though the authors provided a list of enriched taxa in each GM group, bacterial abundance difference between groups is an important factor, and it has to be discussed elaborately in the discussion section.

Authors have only used microbiome sequencing at 9 weeks, however, it is unclear if the difference in body weight after 3 weeks is associated with gut microbiome changes as its 16S seq was not performed at 3 and 6 weeks intervals.

To discuss the influence of the gut microbiome on food intake or body weight, the authors need to provide more mechanistic explanations. The authors only described the results without underlying mechanisms, so it should be better to provide more possible mechanisms and principles for their findings.

Line 287-293: The authors discuss that KEGG gene categories associated with SCFA production positively correlated with birth weight. They should better do a PICRUSt2 analysis to calculate the KEGG gene abundance and compare each group rather than only mentioning butyrogenic taxa in GM1 mice.

The authors do not discuss the reason why the birth weight of pups was different between GM1 and GM4 colonized groups. It needs more explanations with previous findings.

More discussion on sex-specific differences in body weight and microbiome outcomes may be added.

More analyses such as LEfSe, ANCOM and correlational analysis could be done to determine key genera associated with obesity in GM1 vs GM4.

On what basis the authors decide to use 6 and 8 mice and then cull down to 8 mice followed by weaning 20 and 22 mice therefrom? A logical background and power analysis on these numbers should be provided. More information needed on how many males and females were finally analyzed.

How was the microbiome colonization performed? Detailed methodology needs to be provided.

The biggest weakness of the study is the use of only two microbiome sources, which makes the findings very limited in scope, rigor and reproducibility. Why only these sources were chosen? There are many more suppliers and vendors prevalent within the United States that could have been compared in here to make the findings more inclusive.

Another major weakness is the low sample size. It remains unclear if the differences seen in here are exclusively due to supplier-origin microbiome or are also in part due to inherent mouse-to-mouse and cage-to-cage variations in the microbiome. Rigorous power analysis is needed to justify the appropriate number of subjects and to control for such potential underlying biases and confounding factors that otherwise make the data weak and inconclusive.

Minor comments

The authors do not elaborate on some abbreviations (ex. Line 34: SCFAs; Line 62: SPF mice * they elaborate the abbr. at discussion Line 257).

Typos

- Line 109: Number of subtitles (‘2.416. S’ to ‘2.4. 16S)

- Figure 5.B.: They mentioned that they used Bray-Curtis similarities, but probably it is Bray-Curtis dissimilarities

Author Response

We sincerely appreciate your time and effort to review our manuscript.  Based on the comments of all reviewers, the revised manuscript has been greatly improved and we are grateful for your contribution. We list your numbered comments here, followed by our responses to each in bold text. Attached is the revised manuscript, and I am working to upload two new supplemental tables generated based on reviewer suggestions.

The manuscript deals with an interesting topic i.e., the role of vendor-specific microbiomes on fetal growth in outbred mice; however, there are multiple shortcomings that make it unsuitable for publication in the current version.

Mice were colonized with the supplier-origin microbiome, but this is only mentioned in the introduction (Line 75-76), not in Material and Methods.

We appreciate the reviewer’s suggestion and have expanded section 2.2. Mice substantially to include details related to the generation and maintenance of those colonies.

Line 143-144: Authors used SILVA ver. 132 for the taxonomy assignment, but SILVA ver. 138 is available from 2020 and there are huge differences between the two versions. It would be worthwhile to check and update the results with the updated database rather than an outdated and obsolete database.  

We have repeated the annotations using the latest version of SILVA (v138).  The relevant section of the Methods has been revised accordingly.

Figure 1-2: The authors calculated the p-value for the GM and sex difference, but they did not calculate the p-value for the GM difference in the same sex, and sex difference in the same GM.

We performed pairwise comparisons during our original statistical analysis but did not include those results due to the fact that there were no significant GM × sex interactions in any of those outcomes. Based on this and other reviewer comments, we have expanded this section of the Results to provide a more detailed analysis, including p values related to post hoc pairwise comparisons.

Line 175-176: The authors provide results that colonization of GM1 and GM4 generates body weight differences as in the previous study. It has to be discussed precisely in the discussion from the point of view of the microbial content difference of supplier-origin microbiome. To do so, they must measure the microbial composition after colonization and need to link the difference between groups.

Mice in the current study were generated in two separate colonies, and all mice are colonized at birth naturally by their birth dam, with the microbiome (i.e., GM) present in each colony. Thus, there are no ‘pre’ and ‘post’ colonization time-points.  Rather, the current data suggest that the previously reported GM-associated difference in adult body weight is influenced by both fetal growth, as well as intake. The differences in BW demonstrated in the current data were originally considered a confirmatory outcome, but this and other reviewer comments suggest that the taxonomic differences between GM1 and GM4 would be of interest to readers.  Recognizing the differences in output and interpretation of commonly used differential abundance tests, we selected two complementary approaches recently were selected for their consistent and generalizable results across multiple datasets (as shown by Nearing et al, 2022, Nature Communications, 13:342, cited in the revised manuscript), and used in combination to enhance rigor and reproducibility. Table S1 lists those taxa identified as significantly different by both tools, while Table S2 provides a longer list of ASVs identified as significantly abundant by ANCOM-BC via structural zeros. We wanted to provide the comprehensive outputs of these tests, resulting in the decision to opt for a tabular, rather than graphic, display of the data.

Line 189-192: The way of interpreting and describing the results is a bit strange and misleading. The authors can simply describe the results as “When the daily intake was adjusted by weight, there was a large difference between GMs regardless of sex, indicating a strong GM-dependent effect”. However, the meaning is ambiguous due to the use of unnecessary negative sentences like “can no longer be explained…”.

We appreciate (and agree with) the reviewer’s suggestions and have revised this section accordingly.

Figure 3: Post-hoc comparison between time points of each group and between GMs within the same time point should be presented for accurate interpretation and understanding.

The current figure indicates GM-dependent differences within timepoint.   We chose to include this indication, and not indicators of sex- or timepoint-associated differences, as the microbiome is the independent variable of primary interest, and representation of the other differences resulted in overly busy figures.  That being said, we appreciate the reviewer’s suggestion and have expanded this portion of the Results to better explain the timepoint-associated differences within each group (including relevant p values from post hoc comparisons).

Line 242-250: The authors did not provide any results regarding bacterial abundance and colonization results. Even though the authors provided a list of enriched taxa in each GM group, bacterial abundance difference between groups is an important factor, and it has to be discussed elaborately in the discussion section.

As mentioned above, we have expanded the differential abundance (DA) analysis substantially, using two separate DA tools in parallel to identify taxa differing in abundance between GM1 and GM4.  The rationale for the selection of the DA tools used is provided, and results are presented in tabular formats (Table S1 and Table S2).

Authors have only used microbiome sequencing at 9 weeks, however, it is unclear if the difference in body weight after 3 weeks is associated with gut microbiome changes as its 16S seq was not performed at 3 and 6 weeks intervals.

We recognize that this is a limitation of the current study, but consider it a very minor one.  We have sequenced the fecal microbiome of mice from these colonies at ages ranging from one week to over one year (unpublished and Ref 21 in original manuscript).  For all practical intents and purposes, the fecal microbiome of SPF mice is fully established by weaning and very little change occurs over the period of time studied here. 

To discuss the influence of the gut microbiome on food intake or body weight, the authors need to provide more mechanistic explanations. The authors only described the results without underlying mechanisms, so it should be better to provide more possible mechanisms and principles for their findings.

This is an excellent suggestion and we have added text (lines 358 to 367) describing possible mechanisms connecting the microbiome and feeding behavior.

Line 287-293: The authors discuss that KEGG gene categories associated with SCFA production positively correlated with birth weight. They should better do a PICRUSt2 analysis to calculate the KEGG gene abundance and compare each group rather than only mentioning butyrogenic taxa in GM1 mice.

We apologize for the confusion.  The statement cited by the reviewer was in reference to Gough et al. (2021) EBioMedicine, 68:103421, referenced in the preceding sentence.  We have amended the citations accordingly to make this clear to readers.

The authors do not discuss the reason why the birth weight of pups was different between GM1 and GM4 colonized groups. It needs more explanations with previous findings.

This is an interesting, but very broad topic. We have added text in the Discussion relevant to this topic (lines 373 to 376), along with relevant references discussing crosstalk between the gut microbiome and neuroendocrine signaling, including growth, thyroid, and parathyroid hormones, in multiple contexts.

More discussion on sex-specific differences in body weight and microbiome outcomes may be added.

Regarding sex-associated differences in the composition of the microbiota, we applied the same parallel DA analyses used to identify differences between GM1 and GM4, to investigate sex-associated differences. Notably, neither ANCOM-BC or ALDEx2 identified a single taxon as differentially abundant between females and males. This, and the lack of sex-dependent differences in total or BW-adjusted intake, lead us to believe that sex has minimal, if any, effect on the GM-associated influence on intake.  These results are now presented in lines 331 to 333.

More analyses such as LEfSe, ANCOM and correlational analysis could be done to determine key genera associated with obesity in GM1 vs GM4.

Differential abundance (DA) analyses were performed using ALDEx2 and an updated version of ANCOM (ANCOM-BC) which corrects for sample fraction.  This parallel approach was used to account for the variable results generated by different DA tools, and experimental evidence suggesting that these two tools provide the most consistent outcomes across different datasets. This approach and its rationale are provided in the new section 2.9 of the Methods describing the statistical analysis.  Outcomes are provided as Table S1 (taxa identified by both DA tools) and Table S2 (taxa identified by either tool).

On what basis the authors decide to use 6 and 8 mice and then cull down to 8 mice followed by weaning 20 and 22 mice therefrom? A logical background and power analysis on these numbers should be provided. More information needed on how many males and females were finally analyzed.

Culling of litters at birth to 8 mice per litter was done to control for differences in litter size and availability of maternal nutrition and care during the pre-weaning period. Three litters per GM, born at slightly different times, were each culled in this manner, resulting in a total of 24 mice per GM. Due to challenges in sexing neonatal mice however, six of the 48 total mice were sexed incorrectly, ultimately resulting in 21 pairs of sex- and age-matched mice established as cage-pairs at weaning. We did not perform a formal power calculation prior to this study due to a lack of preliminary data, but based our goal of 12 cages per GM on numerous past studies using similar sample sizes. We have expanded section 2.2. Mice of the Methods to clarify these details.

How was the microbiome colonization performed? Detailed methodology needs to be provided.

We apologize for the lack of detail here, and have expanded section 2.2. Mice of the Methods to explain how these colonies of mice were generated, and how they are maintained.

The biggest weakness of the study is the use of only two microbiome sources, which makes the findings very limited in scope, rigor and reproducibility. Why only these sources were chosen? There are many more suppliers and vendors prevalent within the United States that could have been compared in here to make the findings more inclusive.

The reviewer makes an excellent point, and indeed we tested microbiomes originating from only two of the four main suppliers of SPF mice in the U.S., Jackson Laboratory, Taconic, Charles River Labs, and Envigo. As the names GM1 and GM4 imply, our lab previously maintained four separate colonies, each harboring a microbiome originating from those four producers. The microbiomes were numbered GM1 through GM4 based on order of increasing richness, with GM1 and GM4 falling at opposite ends of the first principal coordinate, the other two being intermediate.  In our experience, mice with GM1 (or GM2) were the heaviest and showed the greatest anxiety-related behavior, while GM4 were consistently the leanest, most active, and showed the least anxiety-related behavior. Even before the publication of Ref. 32, we opted to cull the entire colonies harboring GM2 and GM3, to simplify ongoing mechanistic experiments. This relevant background has been added to the Discussion.

Another major weakness is the low sample size. It remains unclear if the differences seen in here are exclusively due to supplier-origin microbiome or are also in part due to inherent mouse-to-mouse and cage-to-cage variations in the microbiome. Rigorous power analysis is needed to justify the appropriate number of subjects and to control for such potential underlying biases and confounding factors that otherwise make the data weak and inconclusive.

While we appreciate the reviewer’s comment, we respectfully contend that the modest low sample size is less relevant in the context of the several significant GM-associated differences, across three different timepoints, each timepoint itself being the mean of three consecutive daily measurements.  The complete consistency of those differences, in the direction matching our hypotheses, makes it incredibly unlikely that the findings are due to random variability and low sample size. I will yield to editorial discretion, but to call the current data weak or inconclusive without a power analysis seems excessive.

Minor comments

The authors do not elaborate on some abbreviations (ex. Line 34: SCFAs; Line 62: SPF mice * they elaborate the abbr. at discussion Line 257).

We appreciate the reviewer’s careful reading, and have added text clarifying the meaning of all abbreviations at first use.

Typos

- Line 109: Number of subtitles (‘2.416. S’ to ‘2.4. 16S)

This was an error in type-setting by the journal and has been rectified.

- Figure 5.B.: They mentioned that they used Bray-Curtis similarities, but probably it is Bray-Curtis dissimilarities

This figure has been revised slightly, and we have changed the legend to use the term Bray-Curtis distance (synonymous with dissimilarity).

Round 2

Reviewer 1 Report

I still worry about some comments before.

Author Response

We have replied to all comments (from all reviewers) to the best of our ability, and are unsure of what other changes are requested.

Reviewer 3 Report

The manuscript was finished. Almost all comments were answered by the authors. Some of the comments were justified refusal. For example, remove drawings that duplicate information (Fig. 1-3). I leave it to the discretion of the editor. The only remark that remains: if the weights of mice in a pair are left on the graphs, then please indicate the average weight of one mouse in the text (simply divide by 2). This is important in order to understand the body weight of just one animal and not be confused with the weight indicated in the figure.

Author Response

We appreciate the reviewer's time and effort.  We have amended the text to include the body weights on a per mouse basis at each timepoint.

Reviewer 4 Report

The manuscript has been revised satisfactorily and can be accepted in the revised version.

Author Response

We appreciated the reviewer's time and efforts.